# The Moderating Role of Family Resilience on the Relationship between COVID-19-Related Psychological Distress and Mental Health among Caregivers of Individuals with Eating Disorders in Post-Pandemic China

**DOI:** 10.3390/ijerph20043417

**Published:** 2023-02-15

**Authors:** Yaohui Wei, Zhiqian Li, Lei Guo, Lei Zhang, Cheng Lian, Chengmei Yuan, Jue Chen

**Affiliations:** Shanghai Mental Health Center, Shanghai Jiao Tong University School of Medicine, Shanghai 200030, China

**Keywords:** COVID-19, caregivers, eating disorders, family resilience, psychological distress, resilience, individual resilience, mental health

## Abstract

COVID-19 has amplified long-standing emotional distress for vulnerable families. While abundant research highlights the importance of resilience under adverse circumstances, little has been undertaken to understand its effectiveness in helping caregivers of individuals with eating disorders (ED) navigate pandemic-related challenges. This paper presents findings of a cross-sectional study investigating the effects of COVID-19-related life disruptions (COLD) and COVID-19-related psychological distress (CORPD) on caregivers’ depression, anxiety and stress, as well as the moderation role of individual resilience (IR) and family resilience (FR) during the post-pandemic period in China. A total of 201 caregivers of individuals experiencing ED participated in our online survey from May 2022 to June 2022. The association between pandemic-related stressors (i.e., COLD and CORPD) and mental health conditions were confirmed. FR moderated the relationship between CORPD and mental health outcomes, while IR independently contributed to low emotional distress. We call for intervention programs strengthening caregivers’ FR and IR, which might benefit both patients and caregivers’ well-being in the post-pandemic period.

## 1. Introduction

Since last year, China has adopted a dynamic, zero-COVID policy in which restrictive measures are implemented if necessary [1]. While this policy has been proven effective in preventing Omicron transmission in certain large cities such as Shanghai and Beijing, some have pointed to the potential disruptions in daily activities such as social services and health services [2]. These measures are particularly detrimental to vulnerable people, including socially marginalized groups and caregivers supporting patients with chronic diseases [3,4]. Moreover, the side effects of COVID-19 have been much heralded in recent years to make this policy more convincing. The fear and anxiety associated with the virus infection can seriously impair daily life and social interactions, cause social avoidance, and induce suspicion [5]. Though the epidemic situation has improved, COVID-19-related psychological distress persists, which suggests that COVID-19 may result in long-lasting mental health problems [5].

Eating disorders (ED) significantly impair patients’ well-being and burden family members with financial and caring issues. In China, parents assume the main responsibility of caring for ED patients. Parents of patients with ED exhibited higher levels of anxiety, depression and stress than those of patients with other psychiatric disorders [6,7]. Parents might be at an increased risk of mental health problems following the various life disruptions and psychological distress associated with the COVID-19 pandemic [8]. Critically, investigating factors that alleviate parents’ psychological distress in response to the pandemic is necessary, because it is not only for their benefit, but also correlates with the outcome of ED [7].

Both individual resilience (IR) and family resilience (FR) have been verified as protective factors against psychological distress [9,10]. IR is considered a healthy coping process whereby individuals could adapt positively despite adversity. Families, as social systems, can be described as “resilient” in ways that may support family adjustment and adaptations during a stressful time [11]. Resilience has been demonstrated to be a culturally and contextually sensitive construct. It is necessary to consider the specific culture context when evaluating the effect of resilience on health outcomes [12,13]. In China, people embrace familial connection, a value rooted in traditional collectivist cultures [14]. Therefore, higher FR may be associated with better psychological adjustment when families are coping with the pandemic.

We are not aware of any studies that have focused on the mental health conditions of parents caring for their ED children during the post-pandemic period and how resilience works in this stressful time. Understanding such relationships may offer insights for more effective interventional programs and policies to support their resilience and mental health in responding to public health emergencies, including pandemics. This is especially important considering that caregivers’ mental health positively affects both the caregivers, children and other family members. Therefore, we aimed to examine the mental health of parents of patients with ED and whether resilience moderates the relationship between different pandemic-related stressors and the mental health of these parents during the period. Specifically, the objectives of the study are to examine (1) the association between different COVID-19-related stressors (i.e., COVID-19-related life disruptions and COVID-19-related psychological distress) and mental health conditions (i.e., depression, anxiety and stress) among parents of patients with ED; and (2) whether IR and FR moderate the relationship of these two factors. We predict that greater COVID-19-related stressors will be associated with worse mental health conditions, and that this association will be moderated by FR and IR (Figure 1).

## 2. Materials and Methods

### 2.1. Participants and Recruitment

Caregivers of patients with ED were recruited for this study, via online advertisements on social media, as well as by being directly approached by researchers. The eligibility of caregivers was based on the following inclusion criteria: caring for a child diagnosed with one of the ED by a psychiatrist, being the ill child’s father or mother, principal caregivers in the family, and having lived with their ill child for the past three months. Caregivers were excluded if there were other members in the family who needed more time and energy to be taken care of.

Cross-sectional data collection began on 5 May 2022 and ended on 10 June 2022. Like previous studies [15,16], respondents were provided with an online survey link that was administered via the WeChat-imbedded Questionnaire Star program. Participants could terminate involvement at any time if they chose. The survey was anonymous, and the confidentiality of information was assured. The research received approval from the Ethics Committee of the Shanghai Mental Health Center (2020-32). All participants signed the informed consent to the study before their enrollment.

A total of 247 participants completed the survey. Of these, 225 caregivers met the inclusion criteria (e.g., a doctor diagnosed their child with ED) and gave their consent to participate. As a quality assurance measure, we included an “attention checking” question. For this question, we asked respondents to select a particular response option and those who failed to select correctly would be excluded from the analysis. A total of 201 participants were included in the final sample after excluding 24 cases showing careless responses.

### 2.2. Measures

#### 2.2.1. Demographics

Sociodemographic information included the caregivers’ age, gender, education level, employment status, marital status, family income, ill children’s diagnosis, age, gender and employment status, etc.

#### 2.2.2. COVID-19 Related Stressors

*COVID-19-related life disruptions (COLD)*. According to existing research [17], we selected seven items identifying common life disruptions caused by pandemics (see Appendix A). These items were developed to assess life disruptions caused by the pandemic during the past three months and caregivers were invited to appraise the degree affected they have felt using a 5-point Likert-type scale, with responses ranging from 0 = “Not at all” to 4 = “Extremely”. A total score of these items was calculated with higher values indicating higher levels of pandemic-related life disruptions.

*COVID-19-related psychological distress scale (CORPD)* [5]. The CORPD, developed and validated by Feng et al. for the population of China, is a 14-item questionnaire assessing pandemic-related anxiety, fear and suspicion. All items on this measure are rated using a 5-point Likert scale (1 = “strongly disagree”, 5 = “strongly agree”). In this study, we used the total score of the CORPD as a measure of COVID-19-related psychological distress. This scale has good reliability and validity in the Chinese population [5].

#### 2.2.3. Individual and Family Resilience

*10-item Connor-Davidson Resilience Scale (CD-RISC10)* [18]. CD-RISC10 is a widely used self-report tool evaluating IR. The scores of each item range from 0 to 4, with higher scores indicating greater IR. The CD-RISC10 has well-documented psychometric properties in the population of China [19].

*Family Resilience Assessment Scale (FRAS-32)* [20]. The FRAS-32, developed and validated by Sixbey et al., is a 32-item questionnaire assessing FR. It was comprised of three subscales: family communication and problem solving (FCPS), utilizing social resources (USR) and maintaining a positive outlook (MPO). The scores of each item range from 1 to 4, with higher scores indicating greater FR. The FRAS-32 has well-documented psychometric properties in the population of China [21].

#### 2.2.4. Depression, Anxiety and Stress

*Patient Health Questionnaire-9 (PHQ-9)* [22]. Depressive symptoms were ascertained using the PHQ-9. Participants indicated the extent to which they had experienced depressive symptoms over the past two weeks on a scale from 0 to 3. The PHQ-9 has good reliability and validity in the Chinese population [23].

*Generalized Anxiety Disorder-7 (GAD-7)* [24]. Anxiety status was ascertained using the GAD-7. Each item was rated on a 4-point Likert scale based on their experience in the past two weeks, with higher values indicating a worse mood state. The GAD-7 has good reliability and validity in the Chinese population [25].

*Perceived Stress Scale-10 (PSS-10)* [26]. PSS-10 is a 10-item scale used to measure the perceived stress level. Scores for several items of PSS-10 were reversed, making scoring consistent across items. The resulting item scores were summed to create a total score, where higher scores indicate greater stress. The PSS-10 has good reliability and validity in the Chinese population [27].

### 2.3. Data Analysis

Sociodemographic and clinical characteristics of groups were analyzed using the descriptive statistics, mean (*M*) and standard deviation (*SD*) for continuous variables and % for categorical variables. Bivariate correlations were examined between study variables. Hypotheses were tested using hierarchical regression analyses where the dependent variable was mental health. In the first step, all regression models were controlled for gender, age, marital status, year of study and income level. In the second step, we added pandemic-related stressors (i.e., CORPD or COLD) to the independent variable list. In the third step, we entered IR and FR as moderating variables and tested the interaction between resilience and stressors on each mental health outcome. When significant interaction effects were obtained, a simple slope analysis [28] was conducted at 1 SD above (i.e., high resilience), and at 1 SD below (i.e., low resilience) the mean values of resilience.

## 3. Results

### 3.1. Descriptive Statistics

Table 1 presents the demographic information of the respondents. A total of 201 participants in the total sample were aged 40 to 73 years, 158 (78.6%) were mothers and 117 (58.2%) had full-time jobs. Children with ED were an average age of 19.55 (SD = 4.11), were primarily students (73.6%) and had anorexia nervosa (51.2%).

### 3.2. Correlational Analysis among Measures

Bivariate correlations among study variables of interest are presented in Table 2. As expected, CORPD and COLD were positively correlated with higher levels of depression, anxiety and stress. In contrast, CD-RISC10 and FRAS-32 and their subscales were negatively correlated with depression, anxiety and stress.

### 3.3. The Effect of Individual Resilience and Family Resilience on Mental Health

The results of the hierarchical regression analysis are presented in Table 3 and Table 4. Hierarchical multiple regression analysis was performed to test the effects of COVID-19-related stressors, and the interaction between COVID-19-related stressors and resilience in predicting mental health. Values of COVID-19-related stressors and resilience were mean-centered to avoid multicollinearity [28].

Regarding the effects of pandemic-related psychological distress on mental health, in step one, the covariates of gender, age, marital status, education and income were entered and significantly predicted anxiety, *F* (7, 193) = 2.47, *p* = 0.019. More years of study (*p* < 0.05) were associated with less depression and anxiety. The addition of CORPD in step two was significant in predicting depression (*β* = 1.83, *p* < 0.001), anxiety (*β* = 1.16, *p* < 0.001) and stress symptoms (*β* = 1.98, *p* < 0.001). In step three, we entered IR, FR and the interaction term between CORPD and resilience into the second model. Both greater IR and FR were positively associated with each of the mental health outcomes. Specifically, respondents who reported higher IR were more likely to have lower depression (*β* = −1.51, *p* < 0.001), anxiety (*β* = −1.47, *p* < 0.001) and stress symptoms (*β* = −2.43, *p* < 0.001). Concerning FR, greater FR predicted less depression (*β* = −1.11, *p* = 0.005), anxiety (*β* = −1.04, *p* = 0.003) and stress symptoms (*β* = −1.22, *p* = 0.003).

Moreover, FR significantly moderated the relationship between CORPD and mental health problems (see Figure 2). Simple slope analyses revealed a significant relationship between CORPD and mental health problems at low FR only (depression, *t* = 5.050, *p* < 0.001; anxiety, *t* = 3.757, *p* < 0.001; stress, *t* = 4.965, *p* < 0.001). When FR was high, the association between CORPD and mental health problems became insignificant (depression, *t* = 1.255, *p* = 0.211; anxiety, *t* = 0.362, *p* = 0.717; stress, *t* = 1.098, *p* = 0.274). In other words, the relationship between CORPD and mental health problems was reduced for individuals with high levels of FR.

Regarding the effects of pandemic-related life disruptions on mental health, the addition of COLD in step two was significant in predicting depression (*β* = 1.27, *p* < 0.001), anxiety (*β* = 1.40, *p* < 0.001) and stress symptoms (*β* = 1.52, *p* < 0.001). In step three, respondents who reported higher IR were more likely to have lower depression (*β* = −1.67, *p* < 0.001), anxiety (*β* = −1.43, *p* < 0.001) and stress symptoms (*β* = −2.57, *p* < 0.001). Concerning FR, greater FR predicted less depression (*β* = −1.07, *p* = 0.008), anxiety (*β* = −1.00, *p* = 0.003) and stress symptoms (*β* = −1.16, *p* = 0.005). However, the interaction terms were not significant, despite the overall model being significant.

## 4. Discussion

In the context of a post-pandemic world, a growing concern is that it might exacerbate preexisting inequalities in physical and mental health among vulnerable groups, especially those living with individuals with chronic disease [29,30]. It is therefore imperative to investigate factors that protect vulnerable families from the pandemic-related stressors. The findings generally supported our hypotheses. ED families may be physically and mentally affected by the pandemic. On the other hand, IR and FR were respectively compensatory and protective factors that contributed to better mental health outcomes.

COVID-19-related stressors were positively associated with mental health problems after controlling for covariates. This confirms our hypothesis that higher levels of COVID-19-related stressors, whether it be COLD or CORPD, are associated with poorer current mental health. Furthermore, caregivers’ education levels were found to correlate significantly with mental health. Caregivers with relatively high educational backgrounds had a lower risk for depression, anxiety and stress. A possible explanation would be that this population might find more self-care resources and gain a better knowledge of the pandemic, all of which might help reduce mental health problems.

Both IR and FR contributed to the low mental health impact of COVID-19-related stressors. Generally, resilience works in three ways to prevent or alter negative outcomes related to risk exposure: compensatory, protective and challenging [31]. Specifically, the compensatory model posits that resilience directly affects outcomes in an opposite direction to a risk factor. The protective model best explains a situation where resilience moderates or reduces the effects of risk on negative outcomes. In the challenge model, coping mechanisms are developed against future stressors as a result of current risk exposure.

In the present study, IR was compensatory in the sense that it was directly and independently associated with a lower risk of depression, anxiety and stress. Caregivers exposed to higher levels of COVID-19-related stressors, such as pandemic-related life disruptions and psychological distress, were more likely to experience mental health problems, but IR could help compensate for the negative effects of pandemic-related stressors. This result was consistent with those found by Friesen et al. [17], who reported that IR did not serve as a moderator but as compensation for such associations in caregivers of autistic individuals.

Interestingly, we found a protective effect of FR on the relationship between CORPD and mental health. For caregivers with higher levels of FR, the positive relationship between CORPD and mental health problems was weaker than that of those with lower levels of FR. These findings echo the protective model of resilience that demonstrates that FR helps to neutralize and weaken the effects of CORPD on depression, anxiety and stress, which implied that intervention with the family system might be effective for caregivers of children with ED. As posited by the Family Adjustment and Adaptation Response model, even in a society where individualism prevails, the capabilities of the family can moderate how individuals experience stressors [32,33]. Especially for caregivers who feel anxious, fearful or suspicious of COVID-19, the effects of pandemic-related psychological distress on their mental health can be detrimental [34]. Inversely, a sense of being able to support family adjustment and adaptations to such challenging experiences emerges as a potential buffering factor against encountering mental health problems. 

In times of large-scale public health crises such as the pandemic, parents are essential in building family resilience. Children may develop resilience from resilient parents, such as how well the parents manage to overcome challenges and take care of their family [35]. Individuals with ED have been through numerous obstacles in the pandemic, such as an increased risks for symptom deterioration and relapse, as well as difficulties in accessing medical care and managing their diet [36,37,38]. Children can adapt better if parents adapt positively during the pandemic. Therefore, interventions that focus on the development of caregivers’ resilience might benefit both patients and caregivers’ well-being. A meta-analysis involving 308 participants reported that psychological interventions, including resilience training, self-disclosure and peer support, may improve psychological resilience among parents of children with cancer [39]. Another meta-analysis supports the effectiveness of resilience interventions combining cognitive behavioral therapy and mindfulness for enhancing psychological resilience, with some evidence of long-term benefits [40]. 

This study has some limitations. The survey is cross-sectional and lacks comparison to a different time point unaffected by the pandemic, thus causal relationships are unable to be established. However, cross-sectional studies are beneficial considering they are relatively cheap, easier, and less time-consuming, especially at this particular point in time [41,42]. In the future, it is necessary to constantly monitor the psychological implications of this population to understand the long-term effect of the pandemic. Second, more detailed confounding variables (e.g., caregivers’ demographic characteristics, severity and course of children’s illness) should be considered in future studies. Third, our data were collected online and relied solely on the self-report of single informant reports from caregivers. Thus, there is a greater chance of common method variance among constructs and responding biases. Fourth, the primary caregivers are composed of fathers or mothers in this study, while other caregivers are excluded, which limits the generalizability of our findings to all caregivers of ED children.

Taken together, our findings provide some empirical evidence to support the employment of resilience frameworks in helping vulnerable caregivers cope with crises like COVID-19. IR and FR are compensatory and protective factors, respectively, in promoting mental health. Resilience can be learned and trained, and, therefore, FR and IR-oriented interventions should be provided to vulnerable caregivers to enhance their ability to rebound successfully post-pandemic. Furthermore, governments should be sensitive to the current challenges for vulnerable families and be flexible in terms of medical resources and supportive policies and practices to help them navigate pandemic-related challenges.

## 5. Conclusions

The COVID-19 pandemic will have long-term impacts on families across the globe. The principal caregivers of Chinese individuals with eating disorders experienced some degree of COVID-19-related stressors and mental health problems. Higher rates of mental health problems have been reported by caregivers of individuals with ED experiencing more COVID-19-related stressors. However, IR and FR contributed to the low mental health impact of COVID-19-related stressors. The association between higher educational backgrounds and better mental health conditions was also confirmed during the post-pandemic period. Furthermore, our findings highlight the role of FR in moderating the relationship between CORPD and depression, anxiety and stress, while IR may independently contribute to low emotional distress. 

Given the presence of mental health problems among caregivers of individuals with ED, increased attention to the needs of this group and delivery of support for both caregivers and individuals with ED is warranted. For example, scientific information and guidance on the epidemic will help families relieve psychological distress and prepare for the anticipated challenges of pandemic-related life disruptions. For healthcare practitioners, resilience-training programs that focus on fostering traits of IR and FR could be incorporated into intervention and counseling practices, which might benefit the caregivers, patients and other family members in post-pandemic China. Future research on the dynamics of pandemic-related stressors, resilience and health status is also urgently needed as a way of supporting caregivers as they strive to meet the needs of individuals diagnosed with ED. Successful functioning for the vulnerable population involves public policies and programs as well as the adequate funding of these to assure access to the resources all citizens enjoy, and which contribute to a better mental health condition.

## Figures and Tables

**Figure 1 ijerph-20-03417-f001:**
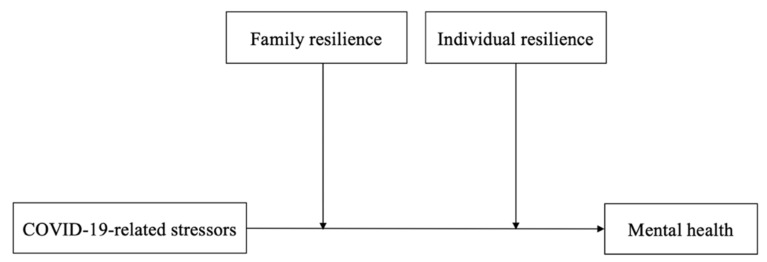
The proposed moderation models. Note: COVID-19-related stressors: COVID-19-related life disruptions and COVID-19-related psychological distress. Mental health: depression, anxiety and stress.

**Figure 2 ijerph-20-03417-f002:**
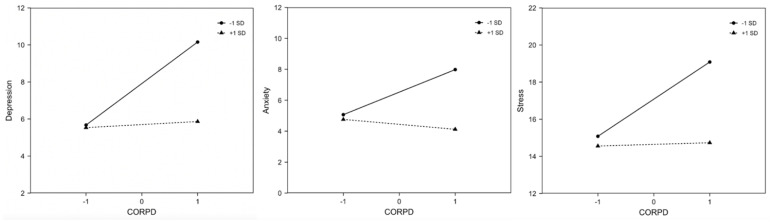
Moderation effect of family resilience between pandemic-related psychological distress and depression, anxiety and stress severity. Note: *SD* = standard deviation; CORPD = COVID-19-related psychological distress scale.

**Table 1 ijerph-20-03417-t001:** Respondents’ demographic information.

Variables	n (%)/M (SD)
Caregiver characteristics	
Gender	
Female (Mother)	158 (78.6)
Male (Father)	43 (21.4)
Age	47.46 (5.03)
Year of study	14.08 (3.35)
Employment	
Full time	117 (58.2)
Other	84 (41.8)
Marital status	
Single parent	23 (11.4)
Two parents	178 (88.6)
Family income (¥)	
5000 below	39 (19.4)
5000–10,000	51 (25.4)
10,000–20,000	64 (31.8)
20,000 or above	47 (23.4)
Child characteristics	
Gender	
Female	194 (96.5)
Male	7 (3.5)
Age	19.55 (4.11)
Employment	
Student	148 (73.6)
Other	53 (26.4)
Diagnosis	
AN	103 (51.2)
BN	61 (30.3)
BED	26 (12.9)
EDNOS	11 (5.5)

Note: *M* = mean; *SD* = standard deviation; AN = anorexia nervosa; BN = bulimia nervosa; BED = binge eating disorder; EDNOS = eating disorder not otherwise specified.

**Table 2 ijerph-20-03417-t002:** Correlations for study variables.

	M (SD)	1	2	3	4	5	6	7	8	9
1. CORPD	36.18 (8.83)	1								
2. COLD	7.77 (4.19)	0.316 **	1							
3. CD-RISC10	26.38 (6.52)	−0.293 **	−0.158 *	1						
4. FRAS-32	93.43 (11.53)	−0.152 *	−0.038	0.463 **	1					
5. FCPS	66.88 (8.83)	−0.136	−0.039	0.439 **	0.988 **	1				
6. USR	8.45 (1.23)	−0.170 *	−0.073	0.348 **	0.666 **	0.583 **	1			
7. MPO	18.10 (2.24)	−0.149 *	−0.004	0.461 **	0.888 **	0.825 **	0.582 **	1		
8. PHQ-9	6.86 (5.28)	0.366 **	0.236 **	−0.444 **	−0.335 **	−0.339 **	−0.241 **	−0.258 **	1	
9. GAD-7	5.47 (4.60)	0.276 **	0.290 **	−0.465 **	−0.358 **	−0.360 **	−0.197 **	−0.316 **	0.818 **	1
10. PSS-10	16.06 (5.79)	0.254 **	0.263 **	−0.566 **	−0.403 **	−0.397 **	−0.305 **	−0.342 **	0.713 **	0.775 **

Note: *M* = mean; *SD* = standard deviation; CORPD = COVID-19-related psychological distress scale; COLD = COVID-19-related life disruptions; CD-RISC10 = 10-item Connor-Davidson Resilience Scale; FRAS-32 = 32-item short version of the Family Resilience Assessment Scale; FCPS = family communication and problem solving; USR = utilizing social resources; MPO = maintaining a positive outlook; PHQ-9 = Patient Health Questionnaire-9; GAD-7 = generalized anxiety disorder-7; PSS-10 = Perceived Stress Scale-10. * *p*-value < 0.05, ** *p*-value < 0.01.

**Table 3 ijerph-20-03417-t003:** Hierarchical regression analyses predicting mental health outcomes.

	Depression			Anxiety			Stress		
	Standardized Coefficient			Standardized Coefficient			Standardized Coefficient		
Predictors	Step 1	Step 2	Step 3	Step 1	Step 2	Step 3	Step 1	Step 2	Step 3
	8.96 *	8.99 *	5.89	5.59	5.61	2.52	17.00 ***	17.03 ***	13.64 ***
Gender	−1.27	−1.04	−0.53	−1.98 *	−1.84 *	−1.34	−1.75	−1.50	−0.65
Age	0.07	0.05	0.09	0.08	0.07	0.11	0.05	0.03	0.06
Marital status	−1.66	−1.21	−0.71	−0.32	−0.04	0.47	−1.46	−0.98	−0.43
Year of study	−0.25 *	−0.23 *	−0.19	−0.23 *	−0.22 *	−0.17	−0.10	−0.08	−0.02
Income 1	0.02	−0.27	−0.58	0.07	−0.11	−0.45	1.07	0.75	0.42
Income 2	−0.20	−0.19	−0.51	0.06	0.07	−0.23	−0.08	−0.07	−0.40
Income 3	−0.74	−0.25	−0.11	−0.56	−0.25	−0.13	−0.75	−0.22	0.00
CORPD		1.83 ***	1.21 ***		1.16 ***	0.58		1.98 ***	1.04 **
FR			−1.11 **			−1.04 **			−1.22 **
IR			1.51 ***			1.47 ***			−2.43 ***
CORPD × FR			**−1.04 ****			**−0.89 ****			**−0.96 ***
CORPD × IR			0.54			0.62			−0.04
Adjusted R2	0.03	0.14	0.29	0.05	0.10	0.29	0.02	0.12	0.37
F	1.92	4.97 ***	7.89 ***	2.47 *	3.84 ***	7.83 ***	1.55	4.43 ***	10.98 ***

Note: CORPD = COVID-19-related psychological distress scale; FR = family resilience; IR = individual resilience. * *p*-value < 0.05, ** *p*-value < 0.01, *** *p*-value < 0.001.

**Table 4 ijerph-20-03417-t004:** Hierarchical regression analyses predicting mental health outcomes.

	Depression			Anxiety			Stress		
	Standardized Coefficient			Standardized Coefficient			Standardized Coefficient		
Predictors	Step 1	Step 2	Step 3	Step 1	Step 2	Step 3	Step 1	Step 2	Step 3
	8.96 *	10.41 *	7.95 *	5.59	7.19 *	5.39	17.00 ***	18.74 ***	15.63 ***
Gender	−1.27	−1.42	−0.68	−1.98 *	−2.15 **	−1.52 *	−1.75	−1.93 *	−0.91
Age	0.07	0.04	0.07	0.08	0.05	0.07	0.05	0.02	0.04
Marital status	−1.66	−1.30	−0.81	−0.32	0.07	0.35	−1.46	−1.03	−0.64
Year of study	−0.25 *	−0.29 *	−0.24 *	−0.23 *	−0.27 **	−0.23 *	−0.10	−0.15	−0.07
Income 1	0.02	−0.20	−0.55	0.07	−0.17	−0.39	1.07	0.81	0.42
Income 2	−0.20	−0.25	−0.52	0.06	0.01	−0.29	−0.08	−0.14	−0.55
Income 3	−0.74	−0.57	−0.36	−0.56	−0.37	−0.23	−0.75	−0.54	−0.29
COLD		1.27 ***	0.94 **		1.40 ***	1.07 ***		1.52 ***	0.95 **
FR			−1.07 **			−1.00 **			−1.16 **
IR			1.67 ***			1.43 ***			−2.57 ***
COLD × FR			0.21			0.44			0.76
COLD × IR			−0.06			−0.44			−0.65
Adjusted R2	0.03	0.08	0.24	0.05	0.13	0.31	0.02	0.08	0.36
F	1.92	3.24 **	6.38 ***	2.47 *	4.86 ***	8.47 ***	1.55	3.19 **	10.49 ***

Note: COLD = COVID-19-related life disruptions; FR = family resilience; IR = individual resilience. * *p*-value < 0.05, ** *p*-value < 0.01, *** *p*-value < 0.001.

## Data Availability

The data are not publicly available due to the privacy of participants.

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
