# Peer review of "The Moderating Role of Family Resilience on the Relationship between COVID-19-Related Psychological Distress and Mental Health among Caregivers of Individuals with Eating Disorders in Post-Pandemic China"

_ijerph, 2023, doi:10.3390/ijerph20043417_

Round 1

Reviewer 1 Report

The present manuscript aims at examining the relationship between COVID-19 related stressors and mental health among caregivers of patients with an ED and to ascertain if both individual and familial resilience moderate this hypothesized relationship. The authors have designed an online survey and have retrieved a sample of 225 participants. The results show that the stress of caregivers linked to pandemics is associated to depression, anxiety and stress. Further, resilience (i.e., individual and familial) correlates with lower uneasiness.

I think that the authors have done a good review of the literature. The methods are clear so that the study could be reproduced. The results are clearly exposed and the discussion is wide. English language is clear and I have not found any mistake.

I do not have any further suggestion for the authors

Author Response

Thank you for your time reviewing our article. We are grateful for your comments and approval.

Reviewer 2 Report

The article deals with the original topic of the emotional burden of caregivers during a pandemic.

The research methods used, the selection of subjects, and the method of conducting the research using the Internet are, in my opinion, correct and raise no objections. The applied methods of statistical analysis deserve to be distinguished. Results presented in a clear and factual way. Conclusions from the research should be more extensive and include generalizations to practical implications.

Author Response

Thank you for your suggestions and comments. We appreciate you! To make the conclusion part more informative and practical, we have enriched our conclusions and offered several practical implications at three levels: policymakers, practitioners, and researchers. Please see conclusions.

Reviewer 3 Report

The article The Moderating Role of Family Resilience on the Relationship between

COVID-19-Related Psychological Distress and Mental Health Among Caregivers of

Individuals with Eating Disorders in Post-Pandemic China is very well written in scientific language. The research problem was very well defined and justified. The purpose of the research was also well defined and the research model was clearly shown. Measurement methods and statistics were chosen correctly. Also, the description of the obtained results is clear and understandable. My only remark concerns the discussion of the results - I suggest referring more to the literature on the subject.

Author Response

Thank you so much for your suggestions and comments. We have carefully read related literature and included six more literature in the revised manuscript including References 29, 30, 32, 33, 39, and 40. Please see discussion.